# Implementing Climate Change Adaptation in Territory Spatial Planning Systems: Challenges and Approaches Based on Practices in Guiyang

**DOI:** 10.3390/ijerph20010490

**Published:** 2022-12-28

**Authors:** Li Zong, Fan Yang, Xinsheng Pei

**Affiliations:** 1Shanghai Tongji Urban Planning & Design Institute Co., Ltd., Guokang Road 38, Shanghai 200092, China; 2Department of Urban Planning, College of Architecture and Urban Planning, Tongji University, Shanghai 200092, China

**Keywords:** climate change, adaptation, territory spatial planning, planning policy, Guiyang, China

## Abstract

Integrating climate change adaptation into spatial planning has become a global goal in the field of spatial planning. Despite the various relevant policies proposed by governments, there is still a lack of common practice in the field of climate change research and territory spatial planning preparation and research in China. In this study, climate change adaptation planning in the territory spatial planning system (TSPS), based upon risk assessment, is explored using downscaled climate change prediction data (derived from CMIP5) and prefectural master territory spatial planning (MTSP) data from Guiyang. The study found that such practices, despite their feasibility, still face systemic challenges given the current planning system in China, e.g., the deficiency of climate change impact data and analyses, the absence of essential planning tools, and the unsuitability of the current planning system for the integration of adaptation actions. We propose corresponding approaches based on our empirical planning experience and discuss prospects for relevant research and planning.

## 1. Introduction

For decades, coping with climate change has been one of the most conspicuous global governance goals, with many re-search efforts and intergovernmental processes executed by numerous institutions and workgroups. Despite the global convergence of climate actions, observed changes in the climate are undeniable at the global scale and are increasingly apparent at regional and local spatial scales [1]. In the 1980s, the ‘greenhouse effect’ and climate change were distinctly identified in the sustainable development consensus, which addressed climate change as a plausible and serious probability [2]. Henceforth, the evidence of climate change has increased significantly. With the continuous growth in global accumulative heat, 2011–2020 became the world’s warmest decade on record [3]. As the severity and complexity of climate change were further recognized by relevant organizations, climate-change-related actions have become a priority in sustainable development agendas.

### 1.1. SDG13: Climate Action and Spatial Planning

In 2015, the United Nations adopted the integrated 17 Sustainable Development Goals (SDGs) as global development goals [4]. Among the Sustainable Development Goals (SDGs), SDG13 (Climate action) displayed no significant correlation with other SDGs [5]. The low integration between SDG13 and other SDGs may make it challenging to create a sustainable development roadmap for countries or regions. The inconsistency between mitigation and adaptation derived from knowledge production and political will may present further challenges to all climate targets in SDG13. Without pertinent planning approaches, the dichotomy of mitigation and adaptation derived from the accumulated research and approaches interpreted by SDG13 will separate climate planning from action, thus restricting the efficiency of incorporated sustainable development actions to reach objectives. As a multi-scale and trans-sector policy tool, spatial planning is appropriate for integrating adaptation and mitigation into one coherent discourse. In order to achieve the goals proclaimed by SDG13 while incorporating with the other SDGs, spatial planning should be utilized by associated organizations as a policy tool in approaches for adaptation [6].

### 1.2. Climate Change Impacts on China

China has long been affected by climate disasters and climate change. According to the UNDRR (United Nations Office for Disaster Risk Reduction), China was the most flooded country between 2000 and 2019, and the population affected by the flood was half the world’s total [7]. Secondly, China’s surface temperature warming rate was ”obviously higher“ than the global average between 1951 and 2020. Thirdly, extreme weather events were also more prevalent, and devastating compared to climatic data from the 20th century. Extreme precipitation, heat, flooding, drought, and sea-level rise are China’s typical and most significant climate change events. Climate change has heavily impacted China, and its significance will only increase in relevant climate scenarios [8].

### 1.3. China’s Climate Change Strategy

In 2007, China’s National Development and Reform Commission (NDRC) published the first National Climate Change Program, and all province development and reform sectors developed their programs in the following years. Despite the long history of China’s participation in climate projects and protocols, its first explicit climate change target was not set until the 12th Five-Year Plan (2011–2015) [9].

In 2014, China’s National Development and Reform Commission (NDRC) released the first national plan on climate change (2014–2020), which identified fundamental principles, policies, and targets for fighting climate change [10]. Although only dozens of prefectures and counties were involved, this was China’s first climate change plan system that demanded a coordinated climate plan and climate change adaptation action program. The national plan was composed under an international domestic balance, synchronized mitigation and adaptation, integrated technological and institutional innovation, and incorporated governmental and societal principles. Moreover, the plan set goals for greenhouse gas (GHG) emissions limits and carbon footprint limits for unit GDP [11]; together with the net-zero commitment, China’s climate change strategy facilitated a quantitative methodology for mitigation. Conversely, the strategy for adaptation only set land use configuration goals in percentage numbers, and the primary adaptation requirements were merely in descriptive text.

In May 2022, China published the National Climate Change Adaptation Strategy 2035 (NCCAS2035). NCCAS2035 is the first planning document at the national level dedicated to adaptation, and set strategic objectives for related departments and regions, along with critical projects and pilot programs in specific areas [12]. In contrast with prior plans, the adaptation demands in the strategy were more detailed and operable.

### 1.4. Established Studies and Backgrounds

In recent years, it has become an increasingly real challenge to integrate climate considerations into spatial planning processes and to use spatial planning as a vehicle for implementing mitigation and adaptation measures [13]. Spatial planning has gained increasing attention in recent years because of its ability to contribute to the successful implementation of climate change adaptation actions in various ways [14]. The incongruity between the proper and actual place of spatial planning in the policy system has long been a concern in climate change response research [15]. Some studies have also analyzed the impact of policy design on the corresponding behavior from the perspective of policy design, spatial planning, and climate change response [16]. In planning practice, as early as 2011, there was already a study that proposed a feasible path to use disaster risk change as a medium for climate change impact adaptation in the spatial planning process [17].

Moreover, some researchers identified priority actions in spatial planning in different regions using climate change impact scenarios. However, due to the lack of knowledge transfer, scale conflicts, and participatory research methods, climate change adaptation research in spatial planning has focused more on macro policies and less on physical planning and practical adaptation actions [18]. In addition, studies on adaptation to climate change impacts in China in related fields are still based on climate change scenarios and status quo data due to the absence of the elements mentioned above [19].

Although these studies can support spatial planning to a certain degree, it is still problematic for spatial planning to closely integrate climate change impact response with small-scale climate change actions in the absence of high-precision climate change scenario data. As mentioned before, the progress of climate change, and effective responses to climate change, especially adaptation, will be an increasing challenge for China. China’s newly established TSPS will be the spatial carrier of various planning spatial elements. Therefore, territory spatial planning should be able to analyze climate change impacts and propose effective response measures at the corresponding scale. However, we have not seen this in China’s existing research and planning work. Therefore, we explored corresponding planning methods based on downscaled climate change prediction data in the context of Guiyang’s territory spatial planning work. Then we analyzed the defects of current supporting mechanisms and relevant frameworks in the TSPS in this regard. Lastly, we put forward targeted suggestions for solutions and discussed prospects for relevant research and planning.

## 2. Reviews of China’s Planning Systems Related to Climate Change

### 2.1. Relevant Planning Systems and the Relational Structure for Climate Adaptation in China’s National Planning Systems

#### 2.1.1. Development Plans

China’s economic and social development plans (Five-Year Plans) are composed and implemented by development and reform committees at corresponding levels of the government. They are considered the supreme and most essential plans in the national planning system, and all other plans should fall in line (Figure 1). In the current national development plan, coping with climate change is an item under the environmental quality improvement section (after the transfer of climate change-related authority from NDRC to the Ministry of Ecology and Environment). Furthermore, the current plan focused on carbon emissions reduction, and the only mention of adaptation was the objective of ”improving adaptation abilities in construction, agriculture, and infrastructure” [20].

#### 2.1.2. Specialized Plans

Various departments at different levels in China’s government have composed specialized plans. The departments in charge generally set specialized plans, objectives, and time frames according to their practical requirements. The plans should be incorporated with other same-level departments’ planning objectives and should conform to development plans and higher-level departments’ plans [21].

Because adaptation was not previously given due importance in the national development plan, the myriad ensuing provincial, prefectural, and county development plans did not take adaptation into adequate consideration in all kinds of specialized planning practices. Adaptation plans have been conducted in select cities for climate-resilience building pilot projects [22]. In these cities, pilot adaptation plans would provide the required materials for plotting development and corresponding specialized plans to enrich adaptation planning.

Theoretically, according to China’s climate change response strategy, all specialized plans should incorporate adaptation actions to a specific arranged extent (Figure 2).However, in actualized planning practice, only carbons emission reduction plans, energy saving plans, and environmental protection plans are explicitly associated with corresponding demands or measures. Moreover, these plans are implemented under the absence of adequate supporting mechanisms, which renders appropriate adaptation actions in these realms impossible [23]. In conclusion, the convergence between specialized plans and adaptation plans demanded by the national strategy has not yet been achieved.

### 2.2. Climate Change Plan

#### 2.2.1. System Structure of Climate Change Plan: Compilation and Implementation

As types of specialized plans, adaptation strategies and frameworks are cooperatively developed by corresponding departments and supervised by the climate change-related authority in the Ministry of Ecology and Environment (MEE). Resembling other specialized planning systems, after the publication of the national strategy, provincial and prefectural plans were published as supporting policies, and these plans shared an identical compositional procedure and structure [24]. The National Leading Group on Addressing Climate Change drafted the national plan under the auspices of NDRC or MEE with contributions from other relevant sectors and personnel, then published it to the associate executive sectors and provincial governments. The climate change plan system was intended to integrate highly incorporated climate actions into the specialized plans under the direct supervision of NDRC or MEE and coordinate it with the Five-Year Plans composed by NDRC. In the departments’ reform in 2018, the climate change response administration moved from NDRC to the Ministry of Ecology and Environment. Corresponding subordinate departments in provincial and prefectural governments also moved from development and reform commissions to the departments of ecology and environment. New provincial and prefectural plans are still in preparation.

The national plan (NCCAS2035) manifested several implementation approaches through the literature, including task responsibility accountability allocation, trans-sector planning convergence, continuous assessment, and financial support in the implementation section. Alongside these ambiguous planning requirements, the plan only defined a few vital quantitative indicators. It did not detail implementation assessment feedback requirements in the implementation section, especially for adaptation measures.

#### 2.2.2. National and Provincial Plan

The national plan underlined infrastructure, water source management, agriculture, forestry, seashore management, ecological vulnerability, public health, disaster risk reduction, and management in adaptation approaches. By stating general planning requirements, the national plan gained increased flexibility while lowering the practicability and accountability of related planning approaches.

Every province (municipality/autonomous region) has an analogous coordinating agency as the state and the same procedure to compose and publish climate change plans. As a result, the expected analogous plan structures could be found in most provincial and prefectural plans. Due to the similarity to the national plan, provincial plans have the same critical components. Experimental adaptation projects allocated by the national plan will be finalized for some selected provinces. While compatible, some provincial plans would use standard indicators with other same-level, epoch-coordinated plans, e.g., the ratio of water-saving irrigation area to all agricultural area in the agricultural plan [25]. With so few planning requirements furthered, provincial plans’ adaptation actions inherited high flexibility from the national plan. In contrast, the planning requirements at the provincial level for more precision and accountability were unsatisfactory.

China’s climate change planning system is still focused on formulating objectives, especially adaptation planning. Without the support of appropriate mechanisms, even if the plan’s objectives are decomposed into various domains, it is still impossible to propose practical planning actions, not to mention effective implementation of the plan. For example, the ”Enhancing the climate resilience of urban and rural construction” section of Hainan’s provincial climate change response plan [26] proposes actively dealing with the heat island effect, rationalizing the layout of functional areas, carrying out climate change impact assessment, and strengthening the protection of tourism resources, but no corresponding planning actions are proposed. This trait is common to national, provincial, and local climate change response plans. In China’s climate planning system, the necessary skills and knowledge to prepare and implement climate adaptation plans are lacking, and a consensus on specific actions has yet to be formed [19]. The central government has formally required provincial and municipal governments to undertake climate change response planning, resulting in superficial plans that cannot be effectively implemented.

#### 2.2.3. Prefectural Plan

At the prefectural level, only the cities in the pilot programs set by the national plan composed climate plans, while most prefectural regions are covered by the general requirements set by provincial plans (Figure 3). China’s coverage of prefectural adaptation plans is deficient compared to the adaptation objectives. Detailed planning requirements are more common in prefectural than provincial plans due to the planning requirements and measures that can be finalized through more local planning processes. Nonetheless, it is insufficient to enable precise and localized adaptation actions in the prefectural plan.

Take Shenzhen’s adaptation plan as an example [27]. Adaptation planning requirements depend more on climate zoning in the prefectural plans, e.g., specialized drainage facility construction requirements adapted for typhoons and extreme precipitation in subtropical areas. Furthermore, as demonstrated in the plan, quantified adaptation planning requirements emerged in construction and production sections, e.g., utility pipe gallery length, renovated water usage percentage, flood diversion channel length, and disaster shelter number. While spatial-related analyses are absent in prefectural plans, precise adaptation actions cannot be realized in the plans. Although the planning committee acknowledged the spatial heterogeneity of climate change impacts in the planning area, they did not utilize spatial tools to interpret the heterogeneity and legislate corresponding adaptation measures through the plan. In the implementation chapter, the plan demanded a trans-sector coordination mechanism led by the city’s Leading Group on Climate Change, Energy Conservation and Emission Reduction, which has not been specified in the institutional details. In addition, the plan proposed establishing implementation metrics, methods, and accountability mechanisms for the adaptation actions, which were only described as general objectives.

It is not uncommon for other published prefectural adaptation plans to exclude the implementation specifications. Furthermore, in the planning processes in the current system, there is not yet an effective channel for engaging stakeholders in sustained dialogue in the policy area of climate change [28]. From this perspective, these established prefectural plans resemble an adaptation strategy rather than an adequately executable plan.

### 2.3. Territory Spatial Planning

#### 2.3.1. System Structure of TSPS

In May 2019, China proclaimed the establishment of the TSPS, a comprehension spatial planning system consisting of five levels (national, provincial, prefectural, county, and township) and three types (master, detailed, and specialized) of plans (Figure 4). Not all the levels fit into the three types of plans, e.g., detailed plans can be deemed as the zoning code for a specific area, which is usually composed by a local government under the provision of its master plan [29]. Meanwhile, inter-regional plans and basin plans are configured under the category of specialized plans. In this system, master plans bear the strategic intent devised by local governments at all levels [30]. These master plans are defined as the cornerstone of the national spatial planning system in terms of spatial policy making.

#### 2.3.2. Climate Change Adaptation in the System

The provincial and prefectural planning guidelines demand climate change risk assessments in the planning procedures, with few detailed requirements in the form of descriptive demand [31,32]. The provincial guidelines took climate change as a background trend for the analysis of spatial development demand. The prefectural guidelines accounted climate change in assessing the uncertainty and instability of future spatial development and provided it as a variable factor for hazard and risk assessments. They also demand that prefectural plans integrate climate change impacts into flood hazard analyses and strengthen countermeasures for sea-level-rising-induced coastal flooding in coastal cities’ territory spatial plans. The guidelines only explicitly stated the demands mentioned above and did not add any procedural or performance specifications for climate change risk assessments. Few published provincial and prefectural territory spatial plans have included detailed climate change risk assessments—and no public edition of these plans has indicated precise planned adaptation actions across planning areas [33,34]. Furthermore, with the absence of guidelines for detailed planning (zoning), the current TSPS only nominally integrated climate change risk assessment. Moreover, the assessment planning action pathway did not appear in the discussion on climate change adaptation in the TSPS.

#### 2.3.3. Objectives for Territory Spatial Planning on Adaptation Set by the National Strategy

NCCAS2035 set the objective of building a national territory adapted to climate change, which demands that the climate should be comprehensively considered in territory spatial planning. Similarly, climate resources, climate change impact, and the corresponding risk assessment should be strengthened and integrated with territorial spatial planning to enrich the One Map of the national territory spatial planning information system. NAACS2035 also set diverse objectives for three types of territory. By 2035, urban space should be composed of climate-resilient cities with lower population and lower socioeconomic and infrastructural climate risk; agricultural space should enhance its climate change adaptation ability to safeguard the supply of agricultural products; green space should protect the environment in it and continuously provide ecosystem services.

Essentially, these adaptation objectives lack detailed guidance on specific territory spatial planning processes, adaptation actions, and planning methods.

## 3. Materials and Methods

### 3.1. Case Study Area and Current Spatial Planning

Guiyang is the capital of Guizhou province, located in the eastern part of the Yunnan–Guizhou Plateau (Figure 5). Its prefectural administration area is the watershed area of the Wu River basin and the Beipan River basin. The topography of Guiyang is composed of middle-altitude plains and hills, and karst landscapes are broadly distributed. On average, Guiyang has abundant annual precipitation; its annual average temperature is 15.3 °C, the maximum temperature is 35.1 °C, and the minimum temperature is −7.3 °C. Guiyang is prone to geological disasters, floods, freezing, and other meteorological disasters.

According to the regional climate model prediction (dynamically downscaled from selected CMIP5 datasets), Guiyang’s climate features will change significantly in the next 30 years in mainstream climate change scenarios; its annual average temperature will increase by 1.4–1.6 °C compared to the baseline climate state (1986–2005) under RCP4.5. The average annual precipitation in the northern part of the prefectural administration area will increase slightly (Figure 6), while the average annual precipitation in the southern part will increase by 60–100 mm.

Guiyang MTSP is still in the process of composition. The land use master plan and the city master plan of Guiyang were both revised in 2017 and currently serve as master plans in the TSPS. These two established plans did not respond to climate change impacts, let alone organize climate change adaptation actions [35,36]. There was no analysis of climate change and planning measures in the land use master plan. The urban master plan only proposed building a comprehensive meteorological monitoring network and a meteorological database to improve the prediction of catastrophic weather, strengthening the foundation of meteorological disaster prevention.

### 3.2. Methodology

By analyzing adaptation planning measures based on existing relevant plans, we accumulated sufficient background information to conduct an adaptation planning practice in the TSPS. For this paper, we analyzed the attributes of China’s territory spatial planning system, climate change plan system, and related plan systems. Firstly, we took the related research on climate change adaptation in spatial planning as the background. Secondly, we analyzed the current adaptation planning of each related element in the TSPS based on our practical experience of climate change adaptation planning attempted in Guiyang Master Territory Spatial Planning (Guiyang MTSP). Thirdly, we analyzed the deficiencies of climate change adaptation planning in the current TSPS based on our experience of climate change adaptation planning in Guiyang. Lastly, we traced the causes of such deficiencies, then proposed a series of improvement suggestions and conducted a series of related discussions (Figure 7).

### 3.3. Adaptation Planning Framework

Given the absence of climate change response in the established plans, in the process of composing the new Guiyang MTSP, the planning team first summarized the common frameworks for climate adaptation planning and assessment. We then composed an ideal framework for implementing climate change adaptation planning and actions into Guiyang MTSP following the requirements of NCCAS 2035. The framework takes climate change impacts and corresponding scenarios as the background for risk assessments, then executes a comprehensive risk assessment and develops planning strategies and supporting measures (Figure 8).

By trialing planning practice, we should have discovered the advantages and disadvantages of TSPS for implementing adaptation approaches, and the links between corresponding phenomena and causes in the planning process. However, during the process, we found that the supportive environment for the Guiyang MTSP still lacks fundamental data to support the analysis of the impact of climate-change-induced meteorological events, convergent tools for translating adaptation strategies into planning actions, and the ability to effectively embed climate change adaptation actions into the TSPS. A detailed planning process assessment cannot be conducted by the planning team based on the extreme insufficiency of climate-change-related resources. After the planning process, we concluded that there were major defects and proposed several approaches for implementing climate change adaptation in the TSPS.

### 3.4. Major Defects in Planning Process

#### 3.4.1. Deficiency of Climate Change Impact Analyses in Planning Areas

During the planning process, data for conducting climate change impact analyses were found to be seriously insufficient. Regarding climate change features, the meteorology departments of Guiyang city and Guizhou province have relatively well-documented meteorological observations and long-term extreme meteorological event observation records. In addition to the observation records, the meteorological department of Guiyang has conducted brief climate analyses using historical meteorological data [37,38,39]. Nevertheless, both departments lack climate change model data to assess climate change impacts in Guiyang under various RCP scenarios. Currently, only the National Climate Center can process climate change model data and provide high horizontal resolution climate change model data for distinct RCP scenarios across China.

On the spatial planning data side, the natural resources department of Guiyang, which oversees the TSPS of Guiyang, cannot collect and provide detailed fundamental data, such as on the topography, drainage network, and river channels of built-up areas required for urban flood risk analysis. Even if the natural resources department knows where and how to collect the relevant data, it still requires additional work from planners to collect the data separately. The insufficiency of data created unnecessary difficulty with conducting the climate change impact assessments for the planning team in the territory spatial planning process.

#### 3.4.2. Absence of Transmission Tools between Adaptation Strategies and Tactics

In terms of planning actions, although the NCCAS2035 set out overall adaptation goals, these objectives were not broken down into planning action requirements and corresponding monitorable indicators. The insufficiency of detailed requirements and fundamental data made it more difficult to connect the climate change impact analysis to matching planning strategies and actions. Take the agrometeorological drought risk analysis conducted for Guiyang as an example. We used the Standardized Precipitation–Evapotranspiration Index (SPEI) as an indicator for the prediction of agrometeorological droughts. This index allows for the determination of drought severity at different timescales, which is essential for the assessment of different responses to drought in different hydrological, environmental, and socioeconomic systems [40]. The assessment revealed that the monthly exceptional drought risk will increase 1.7–1.8-fold by 2035 and 2–3-fold by 2050 under the RCP4.5 and RCP8.5 scenarios (compared to the baseline climate state of 1986–2005). Agrometeorological droughts will pose a severe threat to agricultural production in northern areas (Figure 9). Therefore, through the planning process, the planning team tried to propose relevant measures to increase the capacity of agricultural water facilities and enhance agrometeorological resilience based on climate change assessment and established studies.

However, the lack of a planning objective–strategy–action structure and the insufficiency of available data on agricultural water facilities and irrigation networks prevented a practical quantitative spatial analysis from identifying weaknesses in the existing network. Therefore, developing accurate planning actions for possible climate change risks became impossible. Furthermore, quantitatively monitoring the effectiveness of the planning during its implementation against the adaptation targets and proposing future improvement strategies have also been obstructed by the insufficiency. Due to the severe insufficiency, the planning team could only optimize and integrate the established planning strategies for agricultural water facilities based on the agrometeorological drought risk assessment results (Table 1) [41].

#### 3.4.3. The Inability of Current Spatial Planning System to Integrate Adaptation Actions

Spatial planning objectives and requirements are split into land use and corresponding regulations in the TSPS [42]. All planning elements in the system have exact spatial carriers. The guiding and constraining relationships between various levels of planning are multiplied by these carriers, e.g., both function areas and land use regulation directly take the spatial boundary as the planning object. Land use regulation must meet the constraints set by the function areas. Therefore, if climate change response is to be implemented in the TSPS, the spatial objects and conduction constraint mechanisms of various planning strategies need to be clarified at each level of planning. Even if there is no specific factor transmission relationship, upper-level planning should still set the macro-objectives for lower-level planning objectives.

Following this mechanism, the climate change risk assessment and planning response in Guiyang MTSP should have been executed under the guidance of Guizhou provincial territory spatial planning. However, the provincial planning, which was in process, did not carry out a complete climate change impact assessment at the provincial level, nor did it express the relevant planning objectives in the planning text [43].

## 4. Results

### 4.1. Approach/Objective-Action Framework

For enhancing the ability of the TSPS to implement climate change adaptation, we proposed an approach/objective–action framework corresponding to the defects observed in the trial planning process (Figure 10) based on the experimental planning practice in Guiyang MTSP. Every action in the framework can serve more than one approach/objective, and the potential connections in the framework are not limited to the discussion in this paper.

### 4.2. Efficient Evaluation: Improve the Assessment Accuracy of Climate Change Impacts across TSPS and Relevant Planning Systems

#### 4.2.1. Enhance Cross-Sectoral Coordination with the Meteorological Sector

In the current process of composing and implementing territory spatial planning, cooperation with meteorological departments is primarily limited to obtaining historical meteorological and corresponding hazard data. Meteorological data are primarily used to prevent and control meteorological disasters, and other cascading hazards or environmental impacts related to meteorological events, and the applications are significantly limited to basic analysis. The principal cause of the limitation is that in the traditional “data collection and analysis” model, meteorological data are passed over solely as summarized forms, which are impossible to utilize in spatial analyses.

Moreover, ordinary planners lack experience analyzing detailed meteorological data, and ordinary meteorologists lack the understanding of the spatial planning needs for climate change data and models [44]. Analysis for climate change response requires a combination of planning and climate domain knowledge, and cross-sectoral collaboration in planning efforts can take the form of joint studies and regular discussions. Through the collaboration, the scientific and engineering objectives, analysis methods, and discipline-specific paradigm of climate study and spatial planning will be communicated clearly; the long-term development needs for both realms will be distinguished effectively.

Cross-sector collaborative tools can be divided into levels of data, analysis methods, and research frameworks. The combination of these levels should result in a toolset that fully empowers staff in involved fields. For instance, in terms of data, regional climate information maps based on climate change scenario data can provide planners and policymakers with the necessary basic information without them having to process large amounts of raw model data. Regarding analysis methodologies, the analysis tools based on fixed-source data can be produced jointly by climatologists familiar with the methodologies and regional planners and then packaged for use by a broader range of relevant researchers and engineers. Regarding research frameworks, standard climate change adaptation and spatial planning research frameworks or guidelines at various spatial scales can be developed by experts in related fields and made publicly available to relevant personnel, lowering the knowledge and technical threshold for related analysis work.

#### 4.2.2. Build and Provide Accessible Climate Change Assessment Tools for Planners

The current typical disaster analysis and planning response model applied in TSPS uses meteorological data to identify possible disaster risks, analyze and assess meteorological disaster risks, optimize the spatial planning layout and propose prevention strategies based on the assessment results. This model can effectively analyze disaster risk when the climate background conditions are relatively stable. However, climate change will fundamentally change the climate background conditions, and subsequent risks will emerge beyond the scope of the traditional analysis based on past climate conditions. Therefore, planners need to make it as straightforward as possible to pinpoint the corresponding risks from the climate change context in their planning processes. Providing planners with accessible climate change impact assessment tools is one effective way to address this issue. These tools should be capable of encapsulating the complexity of climate change analyses to interact with spatial planning layouts directly; this approach has already been deployed effectively in some developed regions [45].

### 4.3. Formulating a Strategy Framework for Adaptation in TSPS

#### 4.3.1. Mirror Existing International Experiences

The integration of climate change responses into spatial planning has become a mainstream practice in developed countries/regions that are more severely impacted by climate change, e.g., in California, the state has a comprehensive climate change assessment and a climate change response strategy, and specific requirements for integrating climate change elements in local spatial plans that follow the response strategy [46]. California’s assessment strategy planning–action framework has formed a comprehensive closed-loop, with each grassroots government required to prepare climate change adaptation plans or address climate change in its disaster preparedness planning. In addition, these measures have permeated the spatial planning system by affecting spatial planning elements and have laid the foundation for implementing climate change response strategies in the spatial planning system.

The experiences of these pioneering regions can be helpful for climate response planning in the TSPS at all levels, from legislation and planning preparation to implementation management. Whether in terms of institutions, analysis methods, or adaptation actions, the experiences of pioneering regions can be applied in the TSPS and significantly decrease the cost of climate adaptation planning and action.

#### 4.3.2. Launch Pilot Projects and Summarize Feedback

For a long time, China has adopted various planning and construction pilot practices in the field of climate change adaptation and has achieved some promising results [47]. However, these established pilots are limited to operating within the institutional system of climate change response, and the interface with spatial planning is insufficient. These pilot projects are still limited to several specific projects or industries and do not take a global view of the territory space for climate change adaptation.

In the process of establishing the TSPS, combined with an implementation monitoring system that is compatible with the TSPS, the performance of spatial planning and implementation in the climate change adaptation process should be monitored in a long-term and stable manner, consequently providing empirical evidence for broader climate change adaptation practice. Furthermore, as the TSPS is forming, the early integration of climate change adaptation into the TSPS can provide as much empirical experience as possible for subsequent planning practice while saving trial and error costs.

### 4.4. Policy for Spatial Planning: Integrating Adaptation Actions through Spatial Planning and Corresponding Regulatory Legislation

#### 4.4.1. Develop Adaptation-Associated Land-Use Policy System on Multiple Levels

As mentioned before, the planning elements of the TSPS must be transmitted utilizing spatial objects. In this way, the requirements of higher-level planning can be effectively implemented in lower-level planning through the spatial use arrangement and regulation, which is the core of the TSPS.

In order to transmit climate change adaptation strategies and actions in the TSPS, it is necessary to combine climate change adaptation with the spatial use policies of each level of territory spatial planning and to implement the rules of development, construction, and utilization activities required for climate change adaptation in the land use policies. Climate change adaptation strategies integrated with land use policies are more effective than separate ones. They can also be flexibly adjusted when spatial use patterns change, achieving a deeper integration of adaptation strategies with the TSPS.

#### 4.4.2. Compose Guidelines for Planning Implementation of Climate Change Adaptation Actions

In the current technical standard system of territory spatial planning, only principal requirements are put forward for the planning articles of climate change adaptation. There is a deficiency of requirements for the arrangement and precision of climate change analysis at all levels of planning and an insufficiency of requirements for the hierarchical classification of climate change adaptation planning measures.

If climate change adaptation planning and measures are to be implemented thoroughly and systematically in the TSPS, technical guidelines covering all planning levels and related fields must be formulated before the actual planning composing processes. The technical guidelines may consist of general rules, specific guidance for each level of planning, and requirements for the interface between the various levels of planning. Furthermore, the elements that constitute the technical guidelines should be integrated into the preparation, implementation, and monitoring of territory spatial planning to promote the total integration of climate change adaptation and the TSPS.

## 5. Discussion and Conclusions

### 5.1. Enhance the Pertinence of Territory Spatial Planning

The development of climate change at medium- and long-term time scales and the interactions between related systems that determine the cascade effects caused by climate change are exceedingly uncertain [48]. Therefore, it is difficult to determine a definitive quantified target and criteria for climate change adaptation at the macrolevel.

The prediction and planning of climate change also need to consider all related systems’ interactions. The planning object of territory spatial planning is a specified location/space. Suppose climate change and its impacts are limited to a specific spatial scope. In this case, the complexity of the study can be significantly reduced, and the applicability of the study can be vastly improved [49]. Climate change response in territory spatial plans should be targeted more towards actionable initiatives based on localized studies. The most critical impact factors should be designated for prediction and response based on a comprehensive climate change analysis framework developed by and for the specific location/space.

### 5.2. Advance the Implementation of Territory Spatial Planning

Currently, the TSPS still lacks a strategy to address climate change, much less to guide the development of climate change response actions through detailed planning. Furthermore, there is no universal standard for the cost-performance evaluation methods and standards of various climate change response actions related to spatial planning in the TSPS, which impedes the development of an efficient and practical strategy.

In this context, to cover the deficit, when climate change adaptation is conducted in the TSPS, the effectiveness of various climate adaptation actions should be evaluated in conjunction with localized climate change risk assessments. In the subsequent master plan and detailed planning process, the effectiveness of various climate change adaptation actions should be considered as an essential factor.

### 5.3. Strengthen Stakeholders’ Engagement

Strengthened engagement can be achieved by broadly public participation in the planning process at the macro-scale and combined restrictive and flexible mechanisms at the micro-level.

At the macro scale, adaptation goals set in spatial planning systems are embedded in the overall governance goals at each level of government. The balance between the costs and benefits of adaptation planning is considered in developing development plans and climate adaptation plans. Spatial planning is primarily a policy tool to reflect this balance. The complexity of the objectives, measures, and actors involved in adaptation planning at the macro scale makes it impractical to design a new balanced approach for this mechanism that involves many planning elements. Unlike development plans and other specialized plans, the TSPS has a relatively broader base of public participation procedures. In the master planning process, stakeholders can express their demands and opinions in the corresponding procedures. The benefits and losses of adaptation goals can be fully discussed in the process, and the distribution of corresponding responsibilities can also be recognized. The extent to which the views of all stakeholders are incorporated could be improved. Therefore, in the existing governance system, the balance of responsibility and benefit distribution of adaptation planning at the macro scale depends on optimizing the public participation process.

At the microscale, however, the objectives and actions of adaptation planning actions are more explicit, and the stakeholders involved in each project are more clearly identified. This premise provides a solid basis for establishing a mechanism to allocate responsibility for adaptation planning at the microscale. Allocating climate change-appropriate costs and benefits at the microscale in the TSPS can be accomplished through a system similar to that of volumetric incentives. In the zoning plan, the property rights and profit and loss status of each parcel or each construction project are relatively determined, and the interest subjects within its exogenous influence are relatively straightforward. When the adaptation scheme is implemented, the highly restrictive objectives proposed by the macro-scale planning can be decomposed into the parcels of the zoning plans. The objectives that need to be achieved are the preconditions for developing and using the parcels. Additionally, setting up reward and penalty clauses is a viable path to guide development subjects to reach the optional goals proposed by the adaptation plan in a market-oriented manner.

### 5.4. Limitations

The advance adaptation planning approaches for the TSPS proposed in this paper have not been effectively trialed in the planning practice, and therefore we cannot guarantee the achievement of optimal results for climate change adaptation in the TSPS. Additionally, some of the climate change adaptation strategies and actions planned under spatial planning policies and corresponding planning systems have not been validated by actualized climate change impacts. Even if the planning approaches, strategies, and actions are efficacious, the possibility of a failure to respond to climate change through spatial planning cannot be eliminated.

### 5.5. Conclusions

As a product of the reform that integrates various types of established spatial plans, TSPS has both the authority of being the only spatial plan system and the complexity of including multi-level and multi-disciplinary plans. As one of the SDG goals and the main agenda of human development in the 21st century, climate change response should be thoroughly implemented in the TSPS.

In order to achieve implementation, it is necessary to integrate the concepts, principles, and methods of climate change adaptation into the TSPS; and to make efficient connections and coordination among various stages and specialties of planning. Future reform also needs to promote the implementation of climate change adaptation actions with the authority of territory spatial planning and to avoid the convergency of various strategies and actions being weakened by the complexity of the TSPS.

Although there is still a dearth of climate change adaptation practices in the field of territory spatial planning, more and more research will be conducted in this field as climate change research progresses, climate change adaptation awareness is strengthened, and territory spatial planning matures. Climate change adaptation in the practice of territory spatial planning will also have greater feasibility and effectiveness and will further contribute to the realization of the SDG13.

## Figures and Tables

**Figure 1 ijerph-20-00490-f001:**
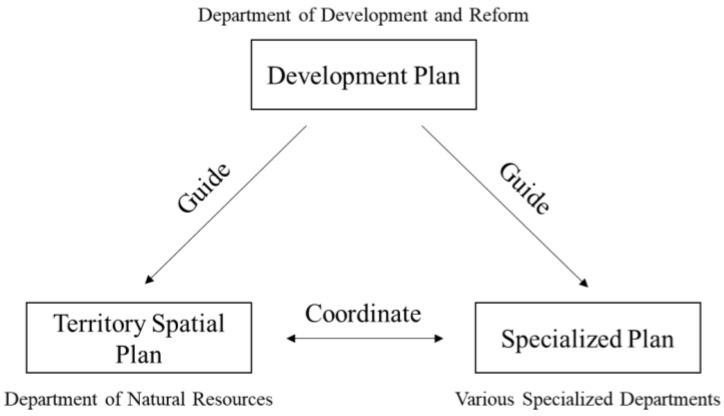
China’s planning system.

**Figure 2 ijerph-20-00490-f002:**
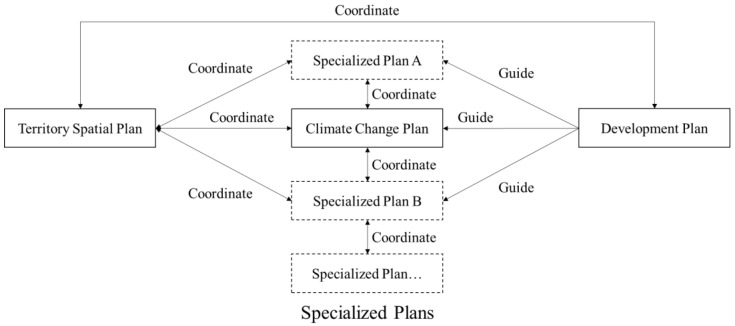
Relationships of the development plan, territory spatial plan and specialized plans on an individual level.

**Figure 3 ijerph-20-00490-f003:**
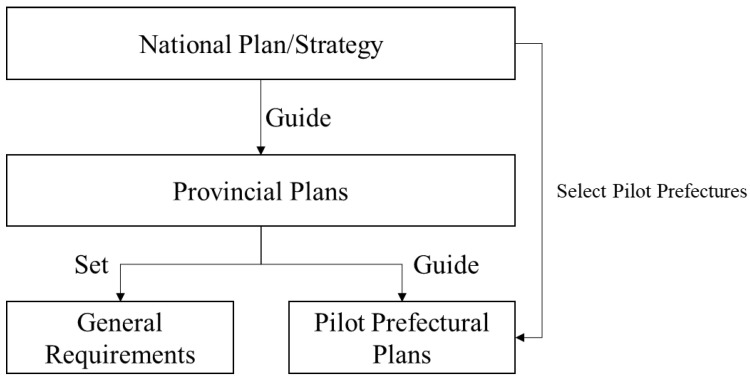
China’s climate strategies and plans system.

**Figure 4 ijerph-20-00490-f004:**
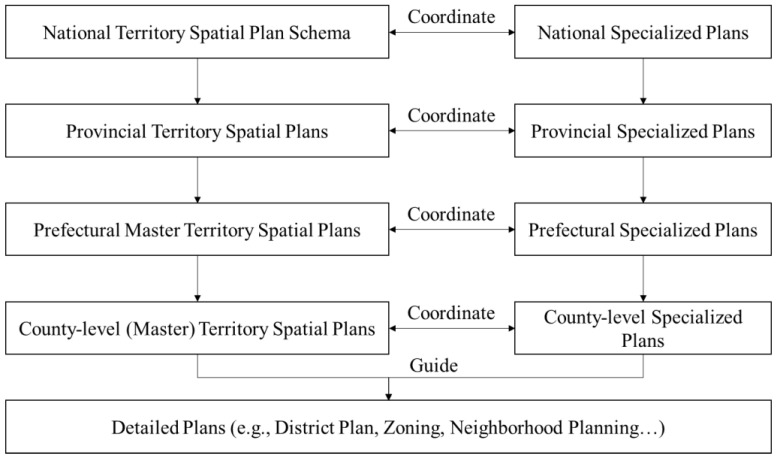
Territory spatial planning system.

**Figure 5 ijerph-20-00490-f005:**
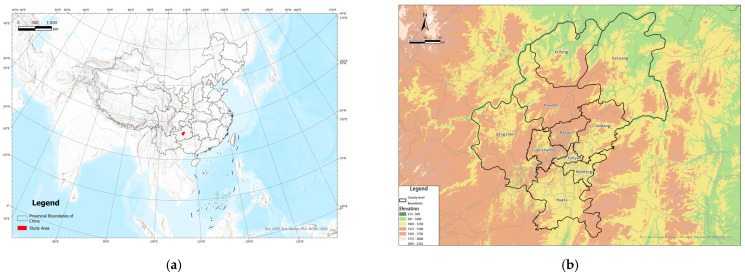
Location of Guiyang. (**a**) Prefectural area. (**b**) Counties of Guiyang.

**Figure 6 ijerph-20-00490-f006:**
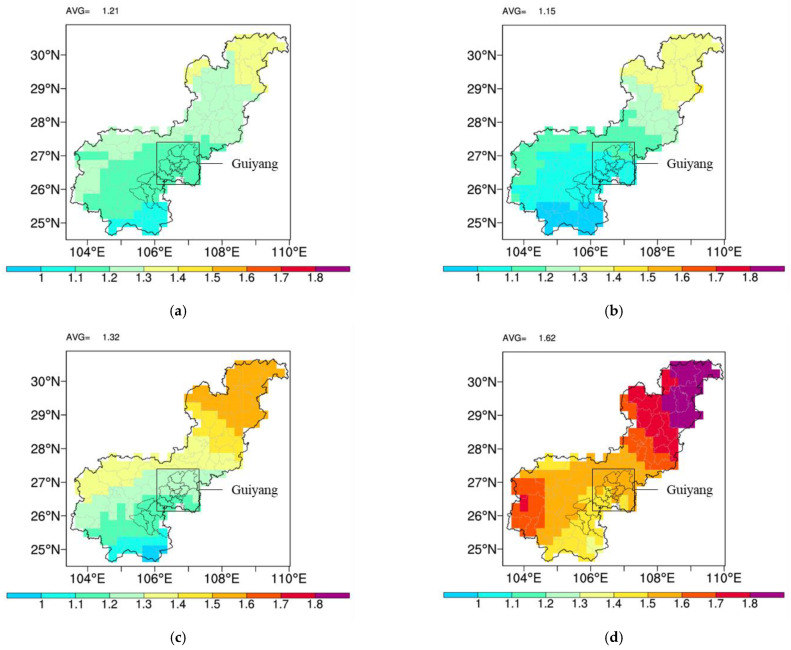
Guiyang–Anshun basin area annual average temperature in 21st century (RCP4.5, 2021–2055 compared to 1986–2005, Celsius Degree). (**a**) 2021–2030; (**b**) 2026–2035; (**c**) 2031–2040; (**d**) 2046–2055.

**Figure 7 ijerph-20-00490-f007:**
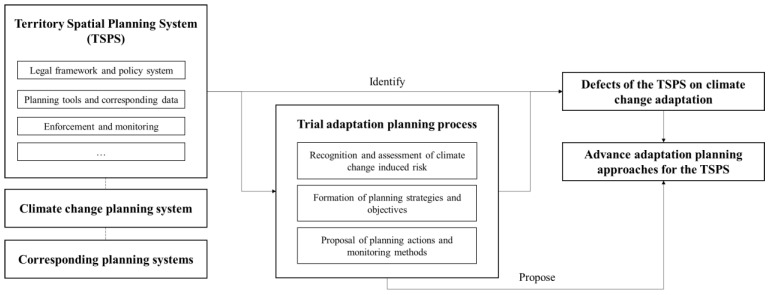
Methodology framework.

**Figure 8 ijerph-20-00490-f008:**
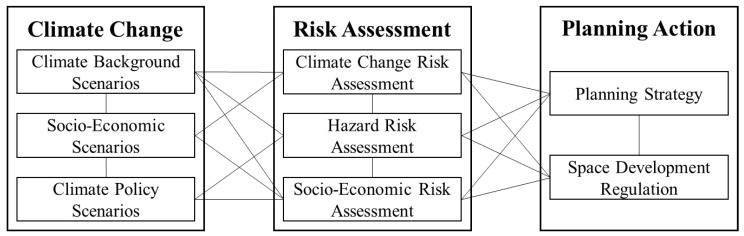
Adaptation framework of Guiyang MTSP.

**Figure 9 ijerph-20-00490-f009:**
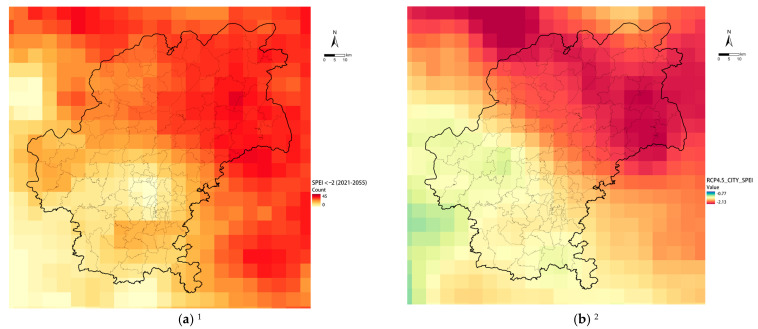
Guiyang agrometeorological drought risk analysis by monthly SPEI under RCP4.5. (**a**) Extreme dry (drought) months count from 2021–2055; (**b**) SPEI distribution in a drought event in 2050s. ^1,2^ Included some area of Anshun (a neighbor prefecture-level city) demanded by the objective of actual planning process.

**Figure 10 ijerph-20-00490-f010:**
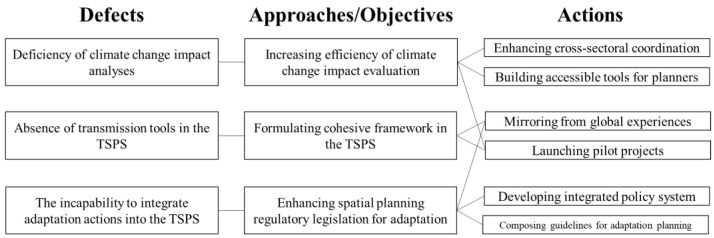
Approach/objective–action framework.

**Table 1 ijerph-20-00490-t001:** Agrometeorological drought adaptation actions proposed in Guiyang MTSP.

Region (County/District/City)	Meteorological Drought Risk Impacts	Major Adaptation Measures
Xifeng, Kaiyang	High	Expand irrigation systems, improve water distribution systems, and water treatment facilities, construct and renovate diversion canals to increase irrigation area and build small water storage facilities ^1^
Xiuwen, Wudang	Medium	Transform sloping farmland to enhance water conservation, conduct ecological restoration, increase vegetation cover ratio to reduce soil erosion
Nanming, Yunyan, Guangshanhu, Huaxi, Baiyun, Qingzhen	Low	Repair water transmission networks to reduce leakage losses, construct water-saving facilities, and improve the water-saving capacity of completed projects

^1^ Based on established analyses and actions.

## Data Availability

Not applicable.

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
