# Peer review of "Implementing Climate Change Adaptation in Territory Spatial Planning Systems: Challenges and Approaches Based on Practices in Guiyang"

_ijerph, 2022, doi:10.3390/ijerph20010490_

Round 1

Reviewer 1 Report

The article is aimed to deepen the study of the effectiveness relationship between overall climate change effects mitigation strategies and spatial planning policies and tools at the different territorial scale level. This is taken as a remarkable case study of Chinese national, regional and provincial policies. Under a general point of view the article tackles an issue of remarkable interest and disciplinary relevance, detecting, although referring to a national case, an overall diffused missing link between general policies  and strategies to deal with climate change and their  effective implementation via spatial/territorial planning policies and tools. 

The case study description is clear and well carried on along with the criticalities that affect the mentioned link at the various levels also in a very structured and institutionally planned system like the studied Chinese context. 

Even though the generals structure of the article is enough clear along with its goals,  in the opinion of the referee some point deserve attention in order to improve the article effectiveness and soundness:

-       The introductory part misses to explicitly describe the methodology approach followed in the article along with a clear description of its structure.

-       References to a, also short, international literature review referred to the studied matter are completely missing, placing the research demand in a disciplinary void, along with a comparison with other research and experiences of possible solutions. Relating to that it is quite strange that reference to the Californian model experience is presented in the final result section

-       Despite the very interesting remarks claiming for a better integration of climate change data analysis in the spatial planning field by devising cross-sector tools suitable to be applied also by planners at the various level, it is not specified or generally contoured the nature and features of such a tools ;

-       The fig. 5 map seems weakly suitable to describe the placement in the national China territory and the geomorphological of the studied area.

Author Response

Thank you very much for your valuable comments.

Reviewer 2 Report

The document presented includes a description on the definition of a Land Master Land Planning in Guiyang. It is an interesting article to be published in a journal of interest to territorial planning professionals, but not at a scientific level. A justification and motivation for the research is not included; besides, the section about methodology does not include a description of the stages, as well as the tools used. The results section has not been obtained by decision-making tools which could have supported the justification of the proposals described. The paper cannot be considered a research article, but not review, because it does not include an analysis of literature review in the field. For all these reasons, the document cannot be accepted for its publication.

Author Response

(The authors gave the same response as above.)

Reviewer 3 Report

The paper presents some important problems with climate change adaptation in China and suggests some remedies. basically, the reasoning is sound though some important elements are missing. The authors point out that the current planning approach lacks a mechanism for sustained stakeholder engagement. This is an important observation. However, this is still missing in the authors' own approach. The paper also lacks discussions about some other important questions including how to distribute responsibility for the different aspects of the adaptation scheme. What is the responsibility of local governments, regional governments, the national government, different agencies, private individuals and commercial actors? There is also a lack of discussion about trade-offs and how to handle them. All adaptation schemes come with a cost, not just in the form of money but also, e.g. in the form of effects on the natural environment, cultural heritage, infrastructure, etc. and all adaptation measures benefit some stakeholders more than others. An adaptation scheme that does not relate to these questions is in practice useless.

Author Response

(The authors gave the same response as above.)

Round 2

Reviewer 3 Report

Over all a big improvement. I only have some minor questions relating to the introduction. The rest of the paper is good as it is.

Introduction: The introduction doesn't really tell me anything about what this paper is about.

"For decades, climate change has been one of the most conspicuous global governance goals," - Climate change can hardly be a goal for anyone. Do you mean climate mitigation? Climate adaptation?

What do you mean by "global convergence"?